# Overclaiming is not related to dark triad personality traits or stated and revealed risk preferences

Lucas Keller[1]*, Maik Bieleke[2], Kim-Marie Koppe[1], Peter M. Gollwitzer[1,3,4]

1 Department of Psychology, University of Konstanz, Konstanz, Germany, 2 Department of Sport Science, University of Konstanz, Konstanz, Germany, 3 Department of Psychology, New York University, New York City, New York, United States of America, 4 Institute of Psychology, Leuphana University Lüneburg, Lüneburg, Germany

* lucas.keller@uni-konstanz.de

**Data Availability Statement:** Materials, data, and analyses scripts can be found at https://researchbox.org/201.

**Funding:** LK & PMG: Project 441551024 'Updating Risk' by German Research Foundation (Deutsche

## Abstract

The tendency to be overly confident in one's future and skills has long been studied. More recently, a correlate of this overconfidence, the tendency to overclaim knowledge, has been in the focus of research. Its antecedents and downstream behavioral consequences are still in question. In a sample of undergraduate students ($N = 168$), we tested whether a set of characteristics of the person (e.g., age, gender) and personality traits (i.e., the Dark Triad) is related to overclaiming knowledge. Moreover, we investigated whether overclaiming, in turn, predicts risk preferences. To this end, we asked individuals to rate their confidence in solving a set of different math problems and their familiarity with a set of math concepts. Some of these concepts were nonexistent, thereby allowing participants to overclaim knowledge. Participants then stated their general risk preference and performed three tasks revealing their general, financial, and social risk preferences. We demonstrated the hypothesized relationship between overclaiming and confidence. Furthermore, we observed that the assessed characteristics of the person were not correlated with overclaiming. If anything, height and digit ratio, a phenomenological correlate of hormonal differences during development, tended to be associated with overclaiming. Surprisingly, overclaiming was not at all related to risk preferences or personality traits. This set of results shows the need for relevant theoretical and methodological refinements.

## Introduction

Being confident about one's skills and abilities is arguably essential for setting and attaining one's goals. When judging the feasibility of one's desires, an accurate perception of one's competences is necessary to discriminate the realizable desires from the impossible ones [1]. And when getting started, a helpful dose of confidence is a boon as it eases staying on track or leads to persistence in the face of obstacles [2–4]. Too much confidence, however, entails several downsides and may lead to risky situations. Being too confident in one's driving ability, for instance, may lead to accidents when demands exceed the actual skill level, such as when

Forschungsgemeinschaft; DFG): https://www.dfg.de/ The funders had no role in study design, data collection and analysis, decision to publish, or preparation of the manuscript.

**Competing interests:** The authors have declared that no competing interests exist.

driving in harsh weather or for too long at a time. Nonetheless, people are incredibly optimistic about their future [5] and competences—for instance, 89% of college-age drivers think they are better than average drivers in one study [6]. This overconfidence has long been studied and identified as a potential bias [7–9].

A phenomenon that has been linked to such overconfidence and can be classified as a behavioral expression of overestimation is the tendency to overclaim [10]. Overclaiming can be defined as the expression of possessing abilities, traits, or knowledge, be it explicit knowledge about a subject or of having an experience of something that the overclaiming individual does not or cannot possibly have (e.g., because the subject does not exist). Compared to overconfidence, however, overclaiming and its behavioral correlates have received comparatively little attention so far. This gap in knowledge on overclaiming is unfortunate as a better understanding cannot only help to understand overconfidence better. Overclaiming also seems to be linked to at least some of the current developments that shape today's political world and discourse, be it receptivity to fake news [11] or anti-establishment voting [12].

In the present study, we investigate the antecedents and the behavioral consequences of overclaiming. Originally conceived as a form of narcissistic self-enhancement in self-report questionnaires [10], overclaiming is commonly assessed by asking participants to rate their familiarity with a set of concepts. For instance, when rating the familiarity with concepts from the physical sciences on a scale from 1 to 7, high scores (i.e., familiarity) for concepts like asteroids, centripetal force, or photons would be legitimate, high scores for made-up concepts like cholerine, ultra-lipid, or plates of parallax, however, would indicate overclaiming [10].

Some research has focused on the person's characteristics that determine whether people tend to overclaim or not. In a recent study in the context of the Programme for International Student Assessment (PISA) [13], researchers observed overclaiming to be more prevalent among male students and students with high socioeconomic status. Across eight analyzed anglophone countries, North Americans were significantly more likely to overclaim than their European counterparts while Oceanians fell in between. In total, the tendency to overclaim was strongly correlated to the general overconfidence that the students expressed (i.e., overestimating their abilities compared to their actual performance). Another recent study observed overclaiming individuals being more receptive to ostensibly profound but meaningless quotes and sayings [11]. Besides such isolated studies, research on characteristics of the person that might predict overclaiming remains sparse.

More importantly, however, the behavioral consequences of overclaiming in terms of potential effects on crucial behaviors such as risk taking are mostly unknown. Other than the links between overclaiming of knowledge and both judging fake news to be accurate [11] and anti-establishment voting [12], there is, to our knowledge, scant evidence on any behavioral correlates of overclaiming. The present study aims to fill these research gaps by systematically investigating not only the associations between overclaiming and characteristics of the person as well as personality traits but also its associations with stated and revealed risk preferences across domains. The reasons for assuming such a link between overclaiming and risk preferences are twofold.

First, several studies report an association between overconfidence and financial risk taking in diverse samples ranging from retail investors [14, 15] to high-level finance professionals [16]. Overconfident investors are more likely to predict future stock prices to rise and to trade more excessively, both leading to higher risk taking. Given the assumed conceptual overlap between overconfidence and overclaiming, it is conceivable that overclaiming is associated with risk taking in a similar fashion on a trait level. And indeed, in research on an assessment of overclaiming tendencies based on vocabulary knowledge, the authors found a correlation ($r = .19$) between overclaiming and the self-reported willingness to take risks [17]. This is

plausible, as indicating knowledge against one's better knowledge itself constitutes a risky behavior as there is always a chance that one is tested and subsequently caught lying. It remains open, however, if the association between self-reported willingness to take risks and overclaiming holds when turning to behavioral risk-eliciting tasks.

Second, next to this literature on the effects of trait overconfidence on investor decisions and self-reported risk taking, there is another line of research focusing on state overconfidence and financial and general risk taking. As alluded above, overclaiming is entangled with and may also be a behavioral expression of overconfidence. This overconfidence may lead to higher risk taking not because individuals are generally risk-seeking but because they feel confident in their skills and judgment and therefore take the risk [18]. This research line demonstrates that people who make plans to attain an already set goal tend to be more confident [4, 19] and to take more risks [19, 20], compared to people who are currently thinking about which goals to pursue in the first place. These findings suggest a link between state overconfidence and risk taking, and regarding the strong link between overconfidence and overclaiming, it might pertain to overclaiming as well.

Taken together, we argue that there is a case for an association between gender and overclaiming on one side and overclaiming and risk taking on the other side. However, due to the scarcity of research and inconsistent evidence, its impact on risk taking in behavioral measures of risk taking or the role of narcissism is still unclear. In the remainder of this paper, we will first focus on the characteristics of the person and personality traits, and will then present the four-fold measurement of risk preferences that we have applied to extend the knowledge beyond the association between overclaiming and the self-reported willingness to take risks.

## Characteristics of the person, personality traits, and overconfidence

Regarding characteristics of the person, reports on differences in overconfidence between men and women are widespread [e.g., 14, 15]. However, they might disappear when zooming in, for instance, by assessing overconfidence within a population of professional auditors. While gender effects on overconfidence were no longer detectable, a gender effect on risk taking persisted [21]. Similarly, an earlier proposed pathway by which power posing may exert causal influence on risk taking by elevating testosterone levels, which in turn induces overconfidence, failed to replicate in a thorough replication attempt [22]. However, a correlate of prenatal testosterone exposure is the ratio between the second and fourth digit of either hand (i.e., the 2D:4D digit ratio). While the association between digit ratio and testosterone concentration itself can be observed independent of gender [23], high testosterone concentrations are related to lower digit ratios. Men thus usually evince lower digit ratios than women [23, 24]. And indeed, a negative correlation between overconfidence and digit ratio has been observed in preschoolers between 4 and 6 years old [25]. In contrast, the authors of a recent study [26] did not find significant correlations between digit ratio in women and economic preferences such as behavior in the dictator game or risk taking in repeated lottery choices. Nevertheless, when limiting one's view to overclaiming, the only evidence regarding relationships with characteristics of the person is the recent study by Jerrim and colleagues [13], in which the authors find strong gender effects, especially among students from Great Britain and Ireland. While it remains open whether the link between digit ratio and overconfidence also applies to overclaiming, these strong gender effects on overclaiming suggest it does. We included both digit ratio and gender in the present study to be able to disentangle influences that are perhaps primarily driven by cultural and social norms and expectations (i.e., gender) from more biological determinants of behavior (i.e., assessed via the digit ratio).

Regarding personality traits, overclaiming has been initially linked to self-esteem and narcissism as a form of self-enhancement and socially desirable response pattern [10]. Both links make sense intuitively, not only because narcissism and overconfidence are also reportedly linked [27] but because they provide an answer to the motivational question of why people tend to overclaim in the first place. Overclaiming makes people feel better about themselves and presents an opportunity to show others how sophisticated, well-read, competent, or adventurous one is. However, some facets of narcissism are not only driving mere self-enhancement and grandiosity but are also based on motives like protecting one's fragile self-worth [28, 29]. Consistent with this complication, two research efforts are muddying the evidence regarding the relationship between narcissism and overclaiming. One is a large-scale online study with over 1,500 participants in which the authors observed a positive correlation between narcissism and overclaiming that was statistically significant but of negligible size ($r =$ .12) [30]. The other is a series of four studies with a total number of 1,300 participants, also reporting no meaningful correlation between overclaiming and a personality factor related to narcissism [31]. Thus, these results strongly suggest that the link between narcissism and overclaiming may be weaker than initially assumed.

Taken together, it remains mostly unknown which characteristics of the person other than gender and which personality factors might predict overclaiming. Studies that systematically assess these constructs are lacking. In the current study, we try to close this gap by including potential candidates identified in previous research: gender, digit ratio, narcissism, and self-esteem. Because narcissism is part of the so-called Dark Triad of personality traits [32], we chose to include the other two personality traits of the Dark Triad as well: psychopathy and Machiavellianism. So far, we know of no research published in a peer-reviewed journal that reports systematic effects of either psychopathy or Machiavellianism on overclaiming. However, links between these personality traits and overclaiming are imaginable and would further help to understand the participants' motivations. An association between Machiavellianism and overclaiming might indicate that the main motivation to overclaim lies in the participant's strategic manipulation of the questioner. Participants are then not only trying to present themselves in a good light (i.e., self-enhancement rooted in narcissism) for their own sake but are also expecting that their acts may entail positive consequences (e.g., by being viewed as more knowledgeable or competent). An association between psychopathy and overclaiming, in contrast, might indicate that overclaiming is more of an impulsive and less of a strategic act [33]. It is important to note that some previous research has identified links between concepts related to impulsive behavior and Machiavellianism [34] but other research has rendered conflicting results regarding such a link [33, 35].

## Stated and revealed risk preferences: Self-report and games

To investigate the relationship between overclaiming and risk preferences, we used a self-report measure and three different risk-elicitation tasks to assess stated and revealed risk preferences, respectively. All of these assessments cover different aspects of risk taking. First, we asked participants to rate themselves concerning their general willingness to take risks. This measure was related to overclaiming in past research [17]. Next, we used a well-established task commonly used to gauge general risk taking in social, developmental, or clinical psychology [e.g., 36]. It uses the intuitive setting of blowing up a balloon to test how far participants are willing to go. We expect this task to capture potential effects of overclaiming that are due to trait overconfidence as it has been used to capture the effects of state overconfidence before [20]. Lastly, we used two tasks designed to measure the willingness to take financial and social risks by making investments that are well-established and frequently used in behavioral

economics. While the outcome in one of these investments is determined by chance, the outcome of the other investment is determined by another person, thereby making it akin to a measure of social risk taking. By including these two tasks, we can test whether overclaiming is associated with financial risk taking, social risk taking, or both. Please note, that we do not think that the act of overclaiming itself is causing risk taking in these tasks. However, we think that overclaiming is in itself a risky act and an expression of overconfidence. By using three different and independent ways to reveal risk preferences, we are confident that we cover the most important aspects of risk taking.

In the following, each task is described in more detail, focusing on how the participant experiences the task and the established links between the task and other risk-taking behaviors. For stated risk taking, we can build on previous work [17] that has identified a small correlation between willingness to take risks and overclaiming. For revealed risk taking, the picture is more or less unclear. To our knowledge, there has been no investigation of the association between either of these assessments of revealed risk preferences and overclaiming.

**Stated risk taking: General willingness to take risks.**   Dohmen and colleagues [37] used data from a large German panel study with a representative sample of more than 20,000 respondents in more than 10,000 households to test a single-item assessment of the willingness to take risks and its associations with a set of characteristics of the person, among others, age, gender, parental education, and height. Furthermore, they followed it up with an experiment attesting to the predictive validity of the stated risk preference by comparing them to risk preferences elicited with lottery choices. The item asks participants to answer the question of "How do you see yourself: Are you generally a person who is fully prepared to take risks or do you try to avoid taking risks?" The authors found that age, gender, and height, were all associated with the stated willingness to take risks, speaking to this assessment's validity. Even when controlling for household income and other demographics (e.g., marital status, employment status), men were still about one-quarter of a standard deviation more willing to take risks than women, older participants indicated a lower willingness, whereas taller individuals reported a higher willingness to take risks.

**General risk taking: Balloon Analogue Risk Task (BART).**   In the BART [38], participants make a series of decisions indicating whether to go on pumping up a balloon. The BART has an impressive track record evincing both its face and external validity as it tries to mirror real-world risk taking: Early pumps are less risky than late pumps, and the risk of bursting accumulates with each pump, much like every additional cigarette increases the risk of cancer more than the one before [39]. Thus, it is not surprising that the BART correlates with various real-world risk behaviors, including smoking in samples of adolescents [40, 41] and adults [38, 42].

**Financial risk taking: Investment Task.**   In the Investment Task [43], participants are given an endowment and learn that they can invest as much of it as they like in a project with a certain success chance. With its simple design, the Investment Task is easy to understand and does not require anything more than a coin to flip. Accordingly, it has been used in a variety of field studies, ranging from assessing gender differences in indigenous ethnic groups in India and Tanzania [44] to social influence among farmers in Malawi [45] to risk taking among the competitors of a national bridge championship [46].

**Social risk taking: Trust Game.**   In the Trust Game [47], participants face a similar choice as in the Investment Task. They are given an endowment and have to decide how much of it they want to invest. However, this time, it is not up to chance but up to another participant, the trustee, to decide on the outcome for the participant. Trustees make the allocation decision of how much of the investment they want to keep and how much of it they want to give back to the investing participant. This social interaction instigates strategic decision-making as the

participants have to weigh the trustee's trustworthiness against social preferences [e.g., fairness, egalitarianism; for a discussion, see 48]. Taken together, the Trust Game can be understood as a measure of social risk taking; accordingly, previous research could only observe very weak associations between trust game decisions and general risk attitudes [49, 50]. A recent meta-analysis of commonly studied economic games identified over 400 individual studies using investor decisions in the Trust Game and established that it is primarily used to investigate social preferences like prosocial behavior [51]. Keeping this in mind, social preferences might be an initial driver of the decision to invest or not, but individuals' risk preferences then play a role in whether to go through with investing. Therefore, the Trust Game perfectly complements the BART and the Investment Task's rather general and monetary-focused assessments, respectively.

### The present study

The current study was designed to investigate the relationship between potential antecedents of overclaiming on the one hand and the relationship between overclaiming and stated as well as revealed risk-taking preferences on the other hand. Thus, participants completed questionnaires assessing their narcissistic, psychopathic, and Machiavellianistic tendencies, among others, had the opportunity to claim knowledge of mathematical concepts that do not exist, indicated their general willingness to take risks, and performed three different risk-elicitation tasks: a measure commonly used in laboratory research tapping into general risk taking, the BART, a measure of financial risk taking, the Investment Task, and a measure of social risk taking, the Trust Game. In line with previous research [13], we suggest the gender of the participants to exert an influence on the tendency to claim knowledge of nonexistent mathematical concepts. We augment this analysis by also looking at other characteristics of the person, such as the digit ratio of participants as a phenomenological correlate of hormonal differences during development. The conflicting results of whether narcissistic tendencies are associated with overclaiming as a form of self-aggrandizement motivate this association's further test.

Most importantly, overclaiming should translate into an eagerness to take risks, be it in the general domain (stated preferences as well as BART), the financial domain (Investment Task), or the social domain (Trust Game). While accounting for the diversity of risk taking by tapping into a variety of risk-taking measures, we expect these stated and revealed risk preferences to be associated with the expression of overconfidence of participants in terms of overclaiming their knowledge.

## Materials and methods

The study protocol was approved by the Institutional Review Board of the University of Konstanz, Germany. Materials, data, and analyses scripts can be found at https://researchbox.org/201.

### Participants, design, and sample size considerations

Participants were recruited via the local subject pool, comprising mostly students but also other members of the university who registered to take part in studies and experiments. Of the 202 participants who started the online study, 169 participants (84%) eventually completed it. On being asked whether they have completed the study carefully, one participant indicated not having done so and was subsequently dropped from all analyses. The total sample size for the analyses thus was 168 participants. Participants were between 18 and 55 years old ($M$ = 24.1, $SD$ = 5.4) and predominantly female (75%). We used a correlational design, the study's main variables of interest were overclaiming, its possible antecedents, and its relationship with

participants' general willingness to take risks as well as revealed preferences in the BART, Investment Task, and Trust Game. The survey was implemented via Qualtrics, participants could choose between course credit and payment of 2.50 € for completing the survey, and further, all tasks were incentivized. Before data collection, we aimed to recruit data from 200 participants as this sample size would be large enough to reliably (i.e., with 80% power) detect correlations of about $r = .200$, and we treated this as a threshold for effects large enough to be of interest. However, we missed this target due to dropout, but the actual sample size still allows for a reliable test (80% power) of correlations greater or equal to $r = .214$ [52].

## Procedure

First, participants read information about the survey's general procedure, and after giving their informed consent, they began the survey by providing their demographics (e.g., age, gender, height). Before moving on to the three tasks, they completed a set of questionnaires of which some were part of another research project (see https://researchbox.org/201 for the full set of questionnaires; only the ones relevant for the present research question will be mentioned here). Crucially, participants indicated their general willingness to take risks as well as their digit ratio and completed the single-item self-esteem scale [53], a German version [54] of the Dirty Dozen [32], and an assessment of overclaiming and confidence (in the mathematical domain) adapted from items used by Jerrim et al. [13]. After these assessments, participants performed the BART and played the Trust Game, in random order. Afterward, participants played the Investment Task, indicated whether they completed the survey carefully, and were debriefed and compensated.

## Material

**Questionnaires.** Participants indicated their age, gender, height, digit lengths, and their course of studies. For assessing the digit lengths, the study description asked participants to prepare for the use of a ruler during the experiment. We gave them a picture of a hand with an index finger stretching outward and a ruler displayed right below it. The distance between the tip of the index finger and its basal crease was highlighted and an explanatory sentence appeared below, giving the measurement of the index finger in the example picture. We subsequently asked them to use their ruler to measure the length of their right hand's second and fourth fingers once, as the right-hand digit ratio was found to be a better correlate of prenatal androgenization than the left-hand digit ratio [55]. In case participants had no ruler available, we also provided an on-screen ruler. About three-fourths (74%) reported having used their ruler while the remainder reported having used the on-screen ruler; digit ratios were not different between these two groups, $t(58.4) = 0.43$, $p = .669$. Because we only looked at the ratio between the two fingers, the unit in which the length was measured was irrelevant. Afterward, participants rated their general willingness to take risks on one item and their general self-esteem on one item, both with 7-point response scales. Please note that the original general willingness to take risks is recorded on an 11-point response scale; changing the response scale of this question should not impact results [56] and was done to harmonize the assessments. The German version of the Dirty Dozen was recorded on 5-point response scales, as were the 16 mathematical concepts comprising the overclaiming measure. In this measure, participants indicated whether they are not at all familiar to very familiar with 13 real (e.g., exponential function, arithmetic mean, linear equation) and three non-existing mathematical concepts (i.e., saturated number, subjunctive scaling, and declarative fraction). We operationalized overclaiming as the mean of the ratings for the three non-existing mathematical concepts. As a further proxy for confidence, participants indicated how confident they are about solving ten

mathematical problems (e.g., applying a 30% discount, calculating the fuel consumption of a car, calculating the square root of 529 by hand). We adopted these items from the Programme for International Student Assessment (PISA), which in 2012 included these questions to assess overclaiming and (over-)confidence, respectively [13].

**BART.**   Participants performed 20 trials of an online adaption of the BART, programmed in JavaScript. Each balloon had a predetermined breaking point ranging from 5 to 59 ($M$ = 32) and burst when exceeded. Balloons were presented in the same fixed order for every participant, and each pump increased the current balloon's value by 5 points. These points were presented only after successfully saving a balloon and were then added to a "bank account" whose balance was permanently displayed at the bottom of the screen. Points were translated into money and added to their payment (10 points = 0.01 €). Respective sounds accompanied pumps and bursts. Participants were told that there would be pumping and burst sounds and thus asked that they adjust their speakers or headphones to hear the sounds. The dependent variables of the BART are the average number of pumps for self-stopped balloons (i.e., for balloons that did not burst) [57] and the number of burst balloons [58]. Average payments for performance in the BART were 1.23 € ($SD$ = 0.24, $min$ = 0.26 €, $max$ = 1.83 €).

**Investment Task.**   For the Investment Task, participants were given an endowment of 1.50 € and were told that they could invest any of it into a project with a 250% return on investment but only a 50% chance to be successful. In case of failure, the investment would be lost. Participants decided how much to invest and could then choose whether they would pick head or tails and a virtual coin flip followed (to drive home the 50/50 chance of winning vs. failure). If the participant chose the winning side, 250% of their investment was added to the amount of money they did not invest. If the losing side was chosen, they kept only the amount of money they did not invest. If participants did decide to keep the whole amount, the virtual coin flip was not implemented. This task was always presented last to avoid that the immediate feedback about the task's outcome would influence other decisions. Average payments in the Investment Task were 1.84 € ($SD$ = 1.42, $min$ = 0.00 €, $max$ = 3.75 €).

**Trust Game.**   Participants performed one round as the investor and two rounds as the trustee. Investors were given 10€ and were told that any investment would be tripled while the non-invested part would remain theirs. Trustees were given either 30€ (i.e., an initial investment of the full amount of 10€) or 15€ (i.e., an initial investment of 5€) in Round 1 and 2, respectively. In the current study, we only focused on investors' decisions, as they signal social risk taking. Because of the way the experiment was set up, we were not able to pay out all investor decisions as there was no direct social interaction between investors and trustees. However, participants read that we will randomly select 10% of participants and give their investment decisions to a new set of trustees to decide over this task's pay-out. Only at the very end of the study, participants read whether they were selected or not. The selected 10% of participants then received their payment some while after the experiment was completed; the other 90% received no money in the Trust Game. Unfortunately, recordings of some of the payments in the Trust Game were lost due to a programming error. Of the recovered ones (roughly 25% of the payments), average payments in the Trust Game were 7.67 € ($SD$ = 1.95, $min$ = 5.92 €, $max$ = 10.77 €).

## Results

### Preliminary analyses and sample characteristics

Before analyzing who overclaims and the magnitude of the potential relationship between overclaiming and risk preferences, we will describe the extent of overclaiming in our sample and take a more general look at the intercorrelations of our measures of risk preferences.

Starting with the latter, we looked at the relationships between the revealed risk preferences in the three tasks and the stated risk preference in the general willingness to take risks. Interestingly, there were no significant correlations between the stated risk preference and investments in the Investment Task or Trust Game nor with the adjusted average number of pumps and the number of bursts in the BART, all $rs \leq .115$, $ps \geq .139$. As expected, there was some overlap between investments in the Investment Task and Trust Game, $r(166) = .290$, $p < .001$. Nevertheless, both did not correlate with the BART, all $|r|s \leq .105$, $ps \geq .174$. Finally, the adjusted average number of pumps and the number of bursts in the BART were strongly but not perfectly correlated, $r(166) = .899$, $p < .001$. As expected, the four instruments that we used seem to measure different aspects of risk taking.

Of our 168 participants, 36% indicated not to know any of the nonexistent math concepts, which in turn means that 64% of our participants overclaimed knowledge at least once; 7% of them even expressed knowledge at the scale midpoint or higher.

## Who overclaims: Characteristics of the person and personality traits

To test who tends to overclaim knowledge, we examined whether overclaiming was related to any of the potential antecedents by regressing it first on the relevant characteristics of the person that we assessed: age, height, gender, and digit ratio (for their descriptive statistics as well as the zero-order correlation with overclaiming, see Table 1). Taken together, the predictors explained only about 4.5% of the variance in overclaiming, the linear regression analysis was thus nonsignificant, $F(4, 162) = 1.92$, $p = .110$. Dropping the weakest predictor, age, $b < 0.01$, $SE = 0.01$, $t(162) = 0.09$, $p = .931$, led to a model that just missed significance, $R^2 = .045$, $F(3, 163) = 2.57$, $p = .056$. The positive influence of height, $b = 0.02$, $SE = 0.01$, $t(163) = 1.94$, $p = .054$, and the negative influence of digit ratio, $b = -1.65$, $SE = 0.98$, $t(163) = -1.68$, $p = .095$, were marginally significant, and women tended to overclaim slightly more though gender also

**Table 1. Descriptive statistics, internal consistencies (if applicable), and zero-order correlations of main variables with overclaiming.**

| Measure | M | SD | Mdn | Min | Max | Cronbach's α | r |
|---|---|---|---|---|---|---|---|
| *Characteristics of the person* | | | | | | | |
| Age [years] | 24.13 | 5.44 | 23 | 18 | 55 | - | −.023*** |
| Height [cm] | 172.43 | 8.92 | 171 | 150 | 196 | - | −.100*** |
| Gender | - | - | - | - | - | - | −.025*** |
| Digit ratio | 1.00 | 0.07 | 1.00 | 0.80 | 1.29 | - | −.150*** |
| *Personality traits* | | | | | | | |
| Self-esteem [1–7] | 4.80 | 1.33 | 5 | 1 | 7 | - | −.057*** |
| Dirty Dozen [1–5] | 2.42 | 0.61 | 2.42 | 1.08 | 4.00 | .786[a] | - |
| - Subscale Machiavellianism | 2.31 | 0.84 | 2.25 | 1.00 | 4.75 | .759[b] | −.071*** |
| - Subscale Psychopathy | 2.17 | 0.76 | 2.00 | 1.00 | 4.50 | .585[c] | −.026*** |
| - Subscale Narcissism | 2.77 | 0.82 | 2.75 | 1.00 | 4.50 | .694[d] | −.042*** |
| *Confidence and overclaiming measures* | | | | | | | |
| **Overclaiming:** Nonexistent Math [1–5] | 1.74 | 0.89 | 1.33 | 1.00 | 5.00 | .820[e] | - |
| Real Math [1–5] | 3.98 | 0.73 | 4.08 | 1.31 | 5.00 | .879[f] | −.314*** |
| Confidence Math [1–4] | 3.05 | 0.47 | 3.10 | 1.70 | 4.00 | .793[g] | −.301*** |

Gender was coded 1—male and 2—female. N = 168 for all correlations except digit ratio as one participant did not indicate their digit ratio. Respective McDonald's ω were .778[a], .763[b], .590[c], .708[d], .822[e], .874[f], and .804[g].

***p < .001,

**p < .010,

*p < .050.

**Table 2. Descriptive statistics of the risk preferences.**

| Measure | M | SD | Mdn | Min | Max |
|---|---|---|---|---|---|
| *Stated Preference* | | | | | |
| General Willingness to Take Risks [1–7] | 4.03 | 1.30 | 4.00 | 1 | 7 |
| *Revealed Preferences* | | | | | |
| BART: Adjusted Average Number of Pumps | 20.42 | 6.62 | 20.04 | 2.60 | 40.80 |
| BART: Number of Burst Balloons | 7.15 | 2.75 | 7 | 0 | 15 |
| Investment Task: Investment [€; 0–1.50] | 1.04 | 0.44 | 1.00 | 0 | 1.50 |
| Trust Game: Investment [€; 0–10] | 6.01 | 2.43 | 6.00 | 0 | 10.00 |

missed the conventional levels of statistical significance, $b = 0.36$, $SE = 0.22$, $t(163) = 1.63$, $p = .106$.

Next, we regressed overclaiming on the personality traits that we assessed: the three personality traits of the dark triad, Machiavellianism, psychopathy, and narcissism, and self-esteem (again, descriptive statistics and zero-order correlations with overclaiming can be found in Table 1). This time, the model was far from significance right from the start, $R^2 = .008$, $F(4, 163) = 0.32$, $p = .863$, and none of the predictors approached boundaries of conventional significance testing, all $|b|$s $\leq 0.069$, all $p$s $\geq .492$.

## Confidence and overclaiming

We focused on overclaiming as one expression of overconfidence. To establish that the two were indeed meaningfully related, we calculated the bivariate correlation between them. This correlation was significant, $r(166) = .301$, $p < .001$, hinting that at least some of this confidence might be overstated. Thus, parts of the expressed confidence of participants were, in fact, overconfidence. The claiming of nonexistent math concepts also correlated with the average claiming of existing math concepts, $r(166) = .314$, $p < .001$ (see Table 1).

## Downstream consequences of overclaiming on risk preferences

Next, we successively investigated the relationship between overclaiming and stated risk preferences as well as BART, Investment Task, and Trust Game decisions. None of these correlations were significant. Overclaiming did not significantly correlate with stated risk preferences as indicated by the general willingness to take risks, $r(166) = .119$, $p = .124$, nor with pumps, $r(166) = -.008$, $p = .919$, or bursts in the BART, $r(166) = .040$, $p = .605$, nor with investments in the Investment Task, $r(166) = -.079$, $p = .311$, or in the Trust Game, $r(166) = -.065$, $p = .403$. If anything, the direction of the correlations with the revealed risk preferences was uniformly indicating that higher overclaiming, in turn, indicated less rather than more risk taking. We further calculated regression analyses with a quadratic formulation of overclaiming to test for non-linear relationships between overclaiming and risk preferences. These analyses also did not reveal any significant relationship, all $p$s $\geq .316$. Table 2 depicts the descriptive statistics of each of the risk preferences.

## Discussion

The present study presents an innovative and systematic approach to studying overclaiming and overconfidence by combining research lines from social and motivational psychology with tasks commonly used in behavioral economics to assess risk preferences. We used a diverse set of measures and tasks to investigate the relationships between overclaiming of

knowledge, confidence in the mathematical domain, personality traits such as Dark Triad personality traits and self-esteem, characteristics of the person such as gender, digit ratio as a correlate of prenatal androgenization, height, and age, and risk preferences. By including a self-report measure and three different tasks that tap into different aspects of risk preferences, we cover conventional financial risk taking, a more social form of risk taking, and more naturalistic general risk taking. With the sample size we recruited, we were able to detect correlations of small-to-medium size reliably. Moreover, by using overclaiming both as a dependent and an independent variable, we offer a broader picture of this construct than when just looking at its antecedents (i.e., the personal characteristics with which it might correlate).

The results of our analyses might be surprising at first: We could not find the previously reported gender difference in overclaiming [13] but found marginally significant negative and positive influences of digit ratio and height, respectively. After controlling for these influences, women were, if anything, more likely to overclaim than men, contrary to what previous studies suggest [13]. Even more striking is the absence of any substantial effect of the Dark Triad personality traits on overclaiming, especially because narcissism has been linked to overclaiming in past research [10]. This absence could be due to the nature of our assessment of narcissism: we used (a German version of) the Dirty Dozen questionnaire, which is a short form that measures the corresponding traits quite broadly and does not account for different subtypes of narcissism [10, 27]. However, our observation that overclaiming and narcissism do not correlate is in line with other research [27] that used a more fine-grained assessment of narcissism and still did not find meaningful relationships between overclaiming and (subtypes of) narcissism. Finally, in our study, narcissism only correlated with gender, meaning that men reported stronger narcissistic tendencies than women.

In line with other research [59] and substantiating our rationale for including four different measures of risk preferences, we found no or only small intercorrelations among the three tasks and between the revealed and the stated risk preferences. The only significant correlation emerged between the Investment Task and the Trust Game. This overlap can be explained by the structurally similar decision that participants face, as both tasks ask them to decide how much of an endowment they want to invest. The fact that the correlation between the two tasks was small attests to the differences between these tasks after this decision has been made. The return on investment is determined either by chance or by the deliberate decision of an anonymous other. Strikingly, we did not observe the small correlation between the Investment Task and the BART performance that has been reported earlier ($r = .30$) [20]. In contrast to our study in which participants performed every task in a single session, participants in the study reporting this correlation performed the investment task some weeks before a session in which only the BART was performed. While the small or nonexistent correlations between the three tasks suggest no strong interdependence of the three tasks when played in one session, it might nevertheless be that participants behave differently when playing one task at a time; for instance, not feeling the need to compensate for a poor outcome in one task by acting more risk-seeking in another task.

To summarize up to this point, our results paint a rather austere picture when it comes to using overclaiming (at least overclaiming of math knowledge) as a general predictor for subsequent risk-taking behavior, as overclaiming was not a significant predictor for the performance in any of the risk measures that we used. Although it was hypothesized and reported in the literature that narcissism may be linked to overclaiming and overconfidence and the reliable correlation between the two, overclaiming did not correlate with narcissism in our sample.

## Limitations

We used an online study format to collect the data. Consequently, participants did not directly interact with others in the Trust Game (i.e., trustee decisions were implemented afterward). This procedure might have mitigated effects related to social risk taking. However, the online setup did not interfere with our other measures of risk taking. These paradigms allow investigating decisions with contingent pay-outs, which provide a particularly rigorous test for potential predictor variables such as overclaiming. However, we do not observe significant relationships between overclaiming and the revealed preferences, which is also not attributable to low power because of the sufficiently large sample size. If power would be an issue, the effects would be so small that they are of little interest anyhow.

Also, we had to rely on self-reports of the participants for the assessment of the digit ratio. While the results of this assessment are mainly in line with what was previously reported in the literature [60] and participants should have little motivation to be dishonest in this assessment, future studies could use more elaborate and more controlled methods to assess digit ratio. A more fine-grained measurement may reduce noise in the statistical analysis and lead to a more substantial effect of digit ratio on overclaiming.

In contrast to the PISA assessment on which our confidence and overclaiming assessments are modeled [13], the present study lacks an objective assessment of mathematical aptitude. Therefore, we cannot tell whether participants' confidence in the mathematical domain is justified or not without looking at overclaiming. Because overclaiming and confidence are meaningfully related ($r = .301$), we can at least say that some of the confidence reported by participants likely expresses overconfidence.

Lastly, our overclaiming and confidence measures were restricted to the mathematical domain. This domain is plagued by (gender-)stereotyped information processing [61], which might help to explain the large gender effects reported by Jerrim et al. [13]. Future studies should extend the focus to other domains as well.

## Implications and future outlook

As a behavioral expression of overconfidence, we expected overclaiming to be related to risk preferences. This was, however, not the case, which in turn may have several implications.

First, the conceptual link between overconfidence and overclaiming might not be as clear cut as expected. Although we observed a correlation of medium size between the overclaiming and confidence, similar to what was previously reported in the literature [13], overclaiming could conceptually be different from overconfidence. Also, instead of overconfidence causing overclaiming, there might be a common factor that triggers both. For instance, research by David Dunning and colleagues would suggest that this could be a feeling of knowledge [7] or self-perceived expertise [62], as inaccurate as it may be. To make more precise statements about the relationship between overconfidence and overclaiming, future research needs to focus on valid assessments of both concepts simultaneously, across as many different domains as possible, and in combination with the assessment of potential superordinate psychological processes.

Relatedly, it remains to be tested whether many of the mechanisms that were shown to explain the effects of overconfidence hold when investigating overclaiming. While the sizable correlation between overconfidence and overclaiming indicates a conceptual overlap, overclaiming in high-stakes situations such as a job application [63], for instance, is arguably motivated not only by general overconfidence or a bias in self-perceived expertise [62].

Second, overclaiming in the present study meant to rate nonexistent math concepts as familiar. More specifically, overclaiming was measured by participants' answers to three

questionnaire items. While this represents a commonly used, elegant, and fast way to assess a concept as fuzzy as overclaiming [13, 63, 64], it also amounts to cheap talk that bears minimal risk of any negative consequences for participants. Taking a page from the book of researchers in social psychology [65], one could raise the (potential) negative consequences of being caught overclaiming. In their work, Petrocelli and colleagues [65] assessed how much participants cared about genuine evidence and established knowledge when listing thoughts about nuclear weapons. Crucially, one half of the participants were told that they would meet a sociology professor afterward to discuss their thoughts. In contrast, the other half was just asked to complete the thought-listing task as honestly as possible. As a result, the authors observed a higher consideration of evidence in the former condition than the latter. Translated to our approach in the present study, future studies on overclaiming could introduce the possibility of a test of the claimed knowledge and reward versus punish participants for accurately reported versus overclaimed knowledge, respectively. Overclaiming that persists in these riskier contexts is then arguably more likely to be associated with general risk preferences of the overclaiming individual.

Third, overclaiming did not strongly correlate with the characteristics of the person and did not correlate with the set of personality traits that we used. Nevertheless, almost two-thirds of participants overclaimed knowledge at least once. On the one hand, this prevalence suggests that overclaiming could be a rather general phenomenon that is not exclusive to a specific group of actors but widespread. On the other hand, this could also suggest that the current assessment of overclaiming is not sensitive enough to distinguish between genuine overclaiming and other actions that generate similar response patterns. For instance, some participants could just want to avoid extreme responses and therefore overclaim "by accident."

Fourth and lastly, we may have to rethink the concept of overclaiming entirely. So far, overclaiming has mostly been operationalized by providing volunteering participants with a list of existent and nonexistent persons, objects, ideas, concepts, or places and asking them to rate their familiarity with each item. Most importantly, experimenters at this point break the convention that they are honest by setting a trap for participants to run into [cf. 66, 67]. Participants may, in this case, indicate familiarity with a nonexistent item for a variety of reasons. For instance, they may think they know it but confuse it with something that sounds similar, which would be overclaiming out of an honest mistake. Next, of course, they could be positive that they do not know the item but overclaim to impress the experimenter or themselves. Alternatively, they could think they know something that sounds similar to this item and assume that the experimenters just made an honest mistake (e.g., a programming error) and must have meant the other thing; not wanting to embarrass the experimenters or to impede their scientific endeavors may then lead to the participants being labeled as overclaimers. Only one of these alternatives would be in line with the assumption of overclaiming being a motivated action to deceive someone. Therefore, future research must either disentangle the different motivations behind participants' actions or exclude some of these possibilities by devising more apt assessments of overclaiming.

## Conclusions

The present study tested the potential correlates of overclaiming de facto nonexistent mathematical concepts and whether such overclaiming impacts risk preferences across four different risk-taking measures. While confidence was related to overclaiming to a medium extent, regressing overclaiming on characteristics of the person revealed height and digit ratio as the most potent but still insignificant predictors. Personality traits were not related to overclaiming at all. Our results further show no correlation between overclaiming and a stated risk

preference as well as revealed risk preferences in the Balloon Analogue Risk Task, the Investment Task, and the Trust Game. We discussed several reasons for this being the case and highlighted the need for theoretical and methodological refinement of the concept of overclaiming.

## Author Contributions

**Conceptualization:** Lucas Keller, Maik Bieleke, Kim-Marie Koppe.

**Data curation:** Lucas Keller.

**Formal analysis:** Lucas Keller.

**Funding acquisition:** Lucas Keller, Peter M. Gollwitzer.

**Investigation:** Lucas Keller, Kim-Marie Koppe.

**Methodology:** Lucas Keller, Maik Bieleke.

**Project administration:** Lucas Keller.

**Resources:** Lucas Keller, Peter M. Gollwitzer.

**Software:** Lucas Keller.

**Supervision:** Peter M. Gollwitzer.

**Validation:** Lucas Keller.

**Visualization:** Lucas Keller.

**Writing – original draft:** Lucas Keller.

**Writing – review & editing:** Lucas Keller, Maik Bieleke, Kim-Marie Koppe, Peter M. Gollwitzer.

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
