## [Decision Letter · Decision Letter 0]

27 Apr 2021

PONE-D-20-37927

Overclaiming is not related to Dark Triad Personality Traits or Stated and Revealed Risk Preferences

PLOS ONE

Dear Dr. Keller,

Thank you for submitting your manuscript to PLOS ONE. After careful consideration, we feel that it has merit but does not fully meet PLOS ONE’s publication criteria as it currently stands. Therefore, we invite you to submit a revised version of the manuscript that addresses the points raised during the review process.

As you can see, both reviewers see that the study has value and contributes some innovative aspects to the literature. I agree with this general evaluation, and I also agree with the reviewers that you should be a bit clearer with respect to the relationship of your personality and behavioral variables. Both reviewers essentially indicate that there are some conceptual overlaps between overclaiming and overconfidence, but this might differ between individuals. The same holds for your dark triad factors. If you could be clearer in your arguments in this part, it would help to understand the contribution of the manuscript.

A second point both reviewers discuss at differing lengths is your use of the measure of prenatal testosterone exposure. While reviewer one has quite some detailed questions about the measure, reviewer two makes a very important point - give a clearer argument why you need that measure at all.

There are also a couple of minor points in both reviews which you might want to consider.

Overall, I think it is important especially in the case of a null-finding to give both a very good interpretation of it and a clear argumentation why an effect could have been expected. It must also be very clear that it is not an artifact of the methods applied or the sample size or other sample characteristics. If this is well done, a null finding is of course also interesting and publishable.

We look forward to receiving your revised manuscript.

Kind regards,

Christiane Schwieren, Dr.

Academic Editor

PLOS ONE

Journal Requirements:

2)  Please note that PLOS ONE uses a single-blind peer review procedure. We would therefore be grateful if you could include in the information that has been redacted for peer review in the manuscript.

3)  Thank you for including your ethics statement:  "Universität Konstanz - Ethics Committee

Bestätigung 15/2019

Written (digital) consent from all participants".   

Please amend your current ethics statement to confirm that your named institutional review board or ethics committee specifically approved this study.

4) We note that you have stated that you will provide repository information for your data at acceptance. Should your manuscript be accepted for publication, we will hold it until you provide the relevant accession numbers or DOIs necessary to access your data. If you wish to make changes to your Data Availability statement, please describe these changes in your cover letter and we will update your Data Availability statement to reflect the information you provide.

Reviewers' comments:

Reviewer's Responses to Questions

**Comments to the Author**

1. Is the manuscript technically sound, and do the data support the conclusions?

Reviewer #1: Yes

Reviewer #2: Partly

2. Has the statistical analysis been performed appropriately and rigorously? 

Reviewer #1: Yes

Reviewer #2: Yes

3. Have the authors made all data underlying the findings in their manuscript fully available?

Reviewer #1: Yes

Reviewer #2: Yes

4. Is the manuscript presented in an intelligible fashion and written in standard English?

Reviewer #1: Yes

Reviewer #2: Yes

5. Review Comments to the Author

Reviewer #1: Summary & general evaluation

On an undergraduate sample, the study examines the associations of overclaiming knowledge with the Dark Triad personality traits (i.e. subclinical forms of narcissism, Machiavellianism, and psychopathy), risk-preference, and other person-characteristics (e.g. gender, 2D4D ratio). The measurement methods applied are diverse including self-reported questionnaires and behavioural measurements. For example, a four-fold behavioural measurement method was used to assess participants’ risk-preference. Overclaiming as the main dependent variable was operationalized with familiarity ratings given on 3 non-existing mathematical concepts. To analyse their data, the authors perform bivariate correlation and multiple regression analyses. The results did not support the hypothesized relationship of overclaiming with Dark Triad tendencies or stated and revealed risk-preferences.

The manuscript is clear, concise, and well-written. The systematic, and broad methodological approach is a high value of the paper. The data is analysed and reported properly. I have only a few comments / questions, please see them below.

Questions & Comments

[1] In the Introduction, the authors present the known associations of overconfidence with personality and behavioural traits and derive their hypotheses about overclaiming from the point that overconfidence and overclaiming are interrelated, although not completely overlapping concepts (“…overclaiming as one expression of overconfidence”, line 336).

From this point, however, it was surprising to me that analyses of associations of overconfidence with the personality and behavioural traits assessed were not performed by the authors. Of course, I understand that the paper is addressed to overclaiming, but considering that the two concepts (overclaiming and overconfidence) are so much related, and that overclaiming had no significant associations with the variables expected, it may help to interpret these negative findings if we see also the associations of overconfidence with the personality/behavioural traits measured in the study.

[2] The authors predict associations of overclaiming with Machiavellianism and psychopathy, mainly based previous findings showing a narcissism-overclaiming association.

I agree that the associations of overclaiming with Machiavellianism and psychopathy are plausible, but the differences between the three Dark Triad traits may require a bit longer introduction to understand the conceptual plausibility of these associations. For example, only a short sentence is provided for the possible association between psychopathy and overclaiming: “An association between psychopathy and overclaiming, in contrast, might indicate overclaiming instead to be more of an impulsive and less of a strategic act. (line 138)” One may expect a longer explanation of this potential association especially because Machiavellianism was also found to be related to self-reported and behavioural impulsivity (i.e. reward sensitivity); see e.g. Birkás, et al. (2015), Personality and Individual Differences, 74, 112-115.

[3] Methods. Digit ratio measurement was used to get an indirect anthropometric indication of the prenatal hormonal environment. The measurement and, also its limitations, are precisely described, only two questions emerged from my part. First, how many times the participants measured the length of their digits? Was it only a single-measurement, or multiple-measurement (e.g. three digit measures, and the mean is used for further analysis)? Second, was there any example image used to explain the correct digit measurement? For example, the length of digits is usually required to be measured from the basal crease of the digits to the tip of the digits. If this is not explained well, then it may be resulted in quite imprecise, and also inconsistent measurements across participants.

[4] Methods. Digit ratio was measured only on the right hand. I suggest the mentioning of the finding that right hand digit ratio is usually a more reliable indicator for testosterone exposure than the left hand ratio. See e.g.: Hönekopp, J. & Watson, S. Meta-analysis of digit ratio 2D:4D shows greater sex difference in the right hand. Am. J. Hum. Biol. 22, 619–630 (2010).

[5] Methods. Sample size calculation is provided, but could the authors give more details about their calculations? For example, how many predictors were used to calculate the a priori power of the regression analysis?

[6] Methods. As a quite valuable part of the study, four-fold measurement of risk preferences (one self-reported measure, and 3 gamified behavioural measures) was assessed. In contrast, however, the most important dependent variable of the study – overclaiming – was tested only by three questionnaire items which specifically related to mathematical problems. I agree that this may be an “elegant and fast way to assess” (line 433) overclaiming, but I feel it a bit a minimalist approach (especially in contrast with the systematic of risk-taking assessment). Could the authors elaborate that how widely this operationalization method was used in previous studies?

[7] Methods. Cronbach’s ɑ for the Dark Triad subscales are rather low. It may be meaningful to show the McDonald's omega in addition to the alpha.

[8] Results and Discussion are quite well written; the authors elaborate many implications of their findings. I have no questions regarding these sections, in this first manuscript version.

Reviewer #2: This is a detailed manuscript reporting the results from an investigation of the relationship between overclaiming one’s knowledge (conceptualized as a facet of overconfidence), gender, personality traits and risk preferences. In order to measure overclaiming experimentally participants were instructed to solve a set of various math problems and their familiarity with a set of math concepts some of them being not existent. Risk preferences were measured by further three behavioral tasks, and digit ratio was included as a measure of testosterone influence during prenatal development. The whole study was performed online. Contrary to their hypothesis overclaiming was found not to be related to risk preferences or the personality traits investigated.

A rather broad introduction focuses on the relevant literature on overconfidence thereby following the assumption of a major conceptual overlap between overconfidence and overclaiming. Notably, although there is a significant correlation between overclaiming and self-confidence in the present study, a correlation of r = .301 does not reflect a high conceptual overlap between both concepts questioning the starting point of this ms. that is “A phenomenon that …can be classified as a behavioral expression of overestimation is the tendency to overclaim” (page 3 line 42/43). The assumption of a high conceptual overlap also constituted the selection of tasks dealing with financial risk taking and personality traits. However, overclaiming may reflect overconfidence in some individuals but may have quite other sources in many individuals as stated by the authors themselves on the top of page of 19.

Some further questions are arising from the selection of tasks and personality measurements:

E.g. although the authors on page 5 cite several studies that suggest no meaningful correlations between overclaiming and narcissism the selection of the Dark Triad as psychometric measurement of personality traits suggest that the authors expected narcissism to be related to overclaiming not considering that narcissism is often based on motives such as hiding one’s fragile self-worth rather than overconfidence.

Page 8: As the authors write, the trust game according to a recent meta-analysis may investigate social preferences like prosocial behavior rather than providing information about social risk taking.

Page 16, 369: Why do the authors control for the influence of digit ratio when they are searching for gender effects?

Page 15: please insert “digit ratio as marker of prenatal testosterone”

I am wondering whether math knowledge and abilities have influenced the results of a very specific and restricted assessment of overclaiming. However, education and knowledge have not been controlled for as far as I see.

In sum, this is a somehow innovative study following an important question which gains special interest in times of a growing amount of fake news. However, I was not surprised that the authors could not confirm their hypotheses since the rationale for their studies is not guided by a stringent understanding of the concepts they intended to investigate.

6. PLOS authors have the option to publish the peer review history of their article (what does this mean?). If published, this will include your full peer review and any attached files.

Reviewer #1: No

Reviewer #2: No

---

## [Author Response · Author response to Decision Letter 0]

4 Jun 2021

Please see the attached file: Response to Reviewers.pdf

---

## [Decision Letter · Decision Letter 1]

13 Jul 2021

Overclaiming is not related to dark triad personality traits or stated and revealed risk preferences

PONE-D-20-37927R1

Dear Dr. Keller,

We’re pleased to inform you that your manuscript has been judged scientifically suitable for publication and will be formally accepted for publication once it meets all outstanding technical requirements.

Kind regards,

Christiane Schwieren, Dr.

Academic Editor

PLOS ONE

Additional Editor Comments (optional):

Reviewers' comments:

Reviewer's Responses to Questions

**Comments to the Author**

1. If the authors have adequately addressed your comments raised in a previous round of review and you feel that this manuscript is now acceptable for publication, you may indicate that here to bypass the “Comments to the Author” section, enter your conflict of interest statement in the “Confidential to Editor” section, and submit your "Accept" recommendation.

Reviewer #1: All comments have been addressed

Reviewer #2: All comments have been addressed

2. Is the manuscript technically sound, and do the data support the conclusions?

Reviewer #1: Yes

Reviewer #2: Yes

3. Has the statistical analysis been performed appropriately and rigorously? 

Reviewer #1: Yes

Reviewer #2: Yes

4. Have the authors made all data underlying the findings in their manuscript fully available?

Reviewer #1: Yes

Reviewer #2: Yes

5. Is the manuscript presented in an intelligible fashion and written in standard English?

Reviewer #1: Yes

Reviewer #2: Yes

6. Review Comments to the Author

Reviewer #1: I carefully read the revised manuscript and the answers given by the authors to my previous questions/comments. The authors were responsive; each of their answers was adequate, and the manuscript was improved by the revision. Therefore, I have no further suggestions, or comments on this manuscript. I suggest an acceptance for publication. Congratulations for the nice contribution to the field!

Reviewer #2: The authors have thoroughly revised the ms. and have considered my critiques in detail. They particularly better defined the constructs they used; so it was important to my perspective to discuss narcissism in its different facets which has now be done.

7. PLOS authors have the option to publish the peer review history of their article (what does this mean?). If published, this will include your full peer review and any attached files.

Reviewer #1: No

Reviewer #2: No

---

## [Editor Report · Acceptance letter]

21 Jul 2021

PONE-D-20-37927R1 

Overclaiming is not related to dark triad personality traits or stated and revealed risk preferences 

Dear Dr. Keller:

I'm pleased to inform you that your manuscript has been deemed suitable for publication in PLOS ONE. Congratulations! Your manuscript is now with our production department. 

Kind regards, 

on behalf of

Dr. Christiane Schwieren 

Academic Editor

PLOS ONE